Data-driven detection of age-related arbitrary monotonic changes in single-cell gene expression distributions

Cheng Jian Hao
Okada Daigo genomeinfoJimu@genome.med.kyoto-u.ac.jp
Center for Genomics Medicine, Graduate School of Medicine, Kyoto University , Kyoto , Kyoto , Japan
Uversky Vladimir
Electronic publication date: 2024 Feb 8
Publication date: 2024
Volume: 12
Electronic Location ID: e16851
Received 2023 Aug 16; Accepted 2024 Jan 8
Copyright: ©2024 Cheng and Okada
Copyright year: 2024
Copyright holder: Cheng and Okada
License: This is an open access article distributed under the terms of the Creative Commons Attribution License, which permits unrestricted use, distribution, reproduction and adaptation in any medium and for any purpose provided that it is properly attributed. For attribution, the original author(s), title, publication source (PeerJ) and either DOI or URL of the article must be cited.
License URL: https://creativecommons.org/licenses/by/4.0/

Keywords: Aging, Computational biology, Single cell genomics, Data analysis method

Funding: KAKENHI Grant-in-Aid from the Japan Society for the Promotion of Science JSPS; Grant No. 21K21316 This work was funded by a KAKENHI Grant-in-Aid from the Japan Society for the Promotion of Science (JSPS; Grant No. 21K21316). The funders had no role in study design, data collection and analysis, decision to publish, or preparation of the manuscript.

==============================
Identification of genes whose expression increases or decreases with age is central to understanding the mechanisms behind aging. Recent scRNA-seq studies have shown that changes in single-cell expression profiles with aging are complex and diverse. In this study, we introduce a novel workflow to detect changes in the distribution of arbitrary monotonic age-related changes in single-cell expression profiles. Since single-cell gene expression profiles can be analyzed as probability distributions, our approach uses information theory to quantify the differences between distributions and employs distance matrices for association analysis. We tested this technique on simulated data and confirmed that potential parameter changes could be detected in a set of probability distributions. Application of the technique to a public scRNA-seq dataset demonstrated its potential utility as a straightforward screening method for identifying aging-related cellular features.

Introduction

Aging is typically associated with a progressive decline in functional integrity and homeostasis, which can give rise to numerous diseases such as heart disease and cancer. Understanding the genetic mechanisms that underlie aging is critical for countering aging- and age-related disease (Barzilai, Cuervo & Austad, 2018; Partridge, Deelen & Slagboom, 2018; Singh et al., 2019). In the field of genetics, considerable effort has been directed toward searching for aging-related genes (Melzer, Pilling & Ferrucci, 2020), and recent omics studies have been conducted to comprehensively identify genes associated with aging in different tissues (White et al., 2015; Kamei et al., 2018; Drummond et al., 2011; Thalacker-Mercer et al., 2010; Su et al., 2015; Patel et al., 2014). In particular, the identification of genes whose expression either increases or decreases with age is central to understanding the mechanisms underlying aging.

Recent scRNA-seq studies have shown that changes in single-cell expression profiles with aging are both complex and multifaceted. Aging is known to be associated with changes in both the number of cells in the population and the gene expression levels in each cell (The Tabula Muris Consortium, 2020). In a study using single-cell technology, age-related increases in the cell-to-cell variability were observed (Yamamoto et al., 2022). Further, cell type-specific aging processes have also been reported for various cell types (The-Tabula-Muris-Consortium, 2020; Wang et al., 2020). While aging-related changes in single-cell expression profiles cannot be detected by bulk-level analysis, single-cell genomics approaches are well suited to detecting age-related changes in single-cell expression profiles.

There have been analyses using transcriptome data from donors of various ages. Studies using bulk data can comprehensively investigate gene expression correlated with age in the target tissue (Tumasian III et al., 2021; Peters et al., 2015). Studies using single-cell RNA-seq will identify cell types whose abundance changes with age and analyze the relationship between donor age and cellular gene expression levels (The-Tabula-Muris-Consortium, 2020). While several methods have been proposed for modeling bulk RNA-seq data that permit sophisticated modeling through linear models (Robinson, McCarthy & Smyth, 2010; Law et al., 2014; Finak et al., 2015), these methods have not yet been adapted to single-cell data. Novel methodological considerations are needed to extend data-driven detection of aging changes in transcriptome data to the entire single-cell expression profile.

In recent years, with the widespread use of single-cell RNA-seq technology, large scRNA-seq datasets have become available. In this study, we developed a novel workflow to detect changes in the distribution of arbitrary monotonic age-related changes in single-cell expression profiles. Since single-cell expression profiles can be analyzed as probability distributions of gene expression, we propose a method that combines the quantification of differences between distributions using information theory, and association analysis based on distance matrices. This technique is potentially useful as a simple screening method for identifying aging-related cellular features.

Methods

Proposed workflow

We developed a novel data analysis workflow to perform an association analysis of the distribution of single-cell gene expression profiles and age (Fig. 1). Our workflow introduces a method that can detect arbitrary monotonic changes in single-cell expression profiles. This is achieved by combining the quantification of distance within probability distributions using information theory, along with an association analysis based on distance matrices. The input data are single-cell RNA-seq data and donor age information from different donors. The single-cell expression data of a gene can be expressed as a probability distribution. By using the distance measure between probability distributions, which has been studied extensively in statistics and information theory, the dissimilarity of the single-cell expression profile between samples can be calculated (Okada et al., 2022).

Figure 1 Graphical illustration of our workflow.

The input data are single-cell RNA-seq datasets from different donors and their donor ages. The single-cell expression data of a gene can be expressed as a probability distribution. By applying distance measures between these probability distributions, which have been studied extensively in statistics and information theory, the dissimilarity of the single-cell expression profile between samples can be calculated. In this workflow, Hellinger Distance, a method widely used to estimate the distance between probability distributions, is used to create a sample × sample distance matrix. A distance matrix based on the absolute value between the donor age of the sample is created. Spearman’s correlation coefficient between the elements of the upper triangular matrix of the distance matrix serves as an indicator of the association between two items based on their distance relationships. This figure was created using Biorender (https://www.biorender.com/).

In this workflow, The following procedure was used to quantify the distance between two probability distributions. First, the maximum and minimum values of the pooled data for each of the two samples were calculated. Between them, 1,000 equally spaced grids were set up, and the probability density of each grid was estimated with the density function of R. The probability density values of all grids were then normalized so that the sum was 1, and the single-cell expression profile was represented by a discrete probability mass function. Hellinger distances were calculated for these probability mass functions. Helligner Distance, a distance metric between two distributions P and Q could be expressed as: (1) HP,Q=12×∑m=1kPm−Qm2

The Hellinger distance is one of the typical distance metrics between two probability distributions and is also used in computational biology methods for cytometry or NGS data (Gingold et al., 2015; Cheng et al., 2023). The advantage of this method is that it can handle arbitrarily shaped probability distributions because it estimates probability distributions and calculates distances in a non-parametric manner.

In addition, an age distance matrix based on the absolute value of the age difference between the two samples was calculated. We used the association analysis between the two distance matrices proposed in a method previously proposed in the field of ecological data science (Somerfield, Clarke & Gorley, 2021). The Spearman coefficient is calculated between the elements of the upper triangular matrix of the distance matrix, which serves as an indicator of this association. Spearman’s correlation coefficient is calculated for each gene to comprehensively examine genes for which changes in the distribution of single-cell expression profiles are associated with aging. The statistical significance of the association was assessed by the Mantel test (Mantel, 1967) between the two distance matrices. In this study, the Mantel test was performed with the default settings for R packages “vegan”(v2.6.4), and the number of permutations is set to 1,000.

Simulation data generation

A simulation data analysis was performed to verify the accuracy of the proposed computational workflow. We established four ordered age categories (Age = 1, 2, 3, 4) and corresponding normal distribution, where the dispersion parameter σ follows three types of the age-related changing pattern (monotonically increasing with age, stabilizing with age, non-monotonic change with age). We considered a total of 20,000 genes, 5% of genes follow the monotonic increasing mode, 5% of genes follow the non-monotonic mode, whereas 90% of genes follow the stable mode. Under the true distribution defined for each age, ten distributions with slightly different means and standard deviations were generated for each age category. Under these true distributions, we generated ten distributions with slightly different mean and standard deviation for each age category.

For each gene, the normal distribution for the i-th sample (Normal(mean = μi, sd = σi)) was generated as follows. The mean (μi) and standard deviation (σi) of the normal distribution for the i-th sample are generated according to the following distribution. μi∼Normalmean=μtrue,sd=0.1

where μtrue is the true mean value and set 10 in this simulation for all ages. σi∼Normalmean=σAge=agei,sd=0.1

where σ[Age = agei] is the true sd assigned to the donor age of the ith sample and agei is the i-th donor’s age. The σ[Age = agei] is assigned different values depending on age. In the gene with monotonic increase pattern, σ[Age = 1], σ[Age = 2], σ[Age = 3], σ[Age = 4] are 0.5, 1, 1.5, 2, respectively. In the gene with a non-monotonic pattern, σ[Age = 1], σ[Age = 2], σ[Age = 3], σ[Age = 4] are 0.5, 1, 1, 0.5, respectively. In the genes with a stable pattern, σ[Age = 1], σ[Age = 2], σ[Age = 3], σ[Age = 4] are all 0.5. This procedure created a set of normal distributions with slightly different parameter values under each age-related parameter changing pattern. From the distribution of each gene of each sample, 1,000 cells were sampled to create statistical samples. We applied the proposed workflow to the simulated dataset to evaluate the performance of the methodology.

Real scRNA-seq data analysis

We then applied our method to the large-scale single-cell RNA-seq dataset for the aging study. We used a public dataset of aging mouse single-cell RNA-seq data (The-Tabula-Muris-Consortium, 2020). We downloaded the preprocessed data file (tabula-muris-senis-droplet-processed-official-annotations.h5ad) from the Tabula Muris Senis website and used it for downstream analysis. This dataset contains single-cell gene expression data for 20,138 genes in 245,389 cells derived from various organs of multiple mouse donors of different ages. The donor mice are 23 individuals with ages of 1 month, 3 months, 6 months, 18 months, 24 months, or 30 months. Of all cells, 164,027 are from male mice and 81,362 are from female mice. Since sex differences are known to occur in age-related changes in gene expression (Boheler et al., 2003; Hägg & Jylhävä, 2021), we only used a male dataset for the downstream analysis. The top 3,000 genes with the highest average expression in the cells for each age category were included in the analysis. We focused our analysis on four organs (kidney, limb muscle, lung, and marrow) that were measured in more than 10 mice in this dataset. We applied the proposed workflow to this preprocessed dataset and calculated the score of the age-related change of the single-cell expression profile for each gene.

Results

Simulation data analysis

A simulation data analysis was performed to verify the accuracy of the proposed computational workflow. We established four ordered age categories (Age = 1, 2, 3, 4) and corresponding normal distribution, where the dispersion parameter σ follows three types of age-related changing pattern (monotonically increasing with age, stable to age, non-monotonic change with age). We considered a total of 20,000 genes, 5% of genes follow the monotonic increasing pattern, 5% of genes follow the non-monotonic pattern, whereas 90% of genes follow the stable pattern. The true distribution plots of each pattern are shown in Fig. 2. Under the true distribution defined for each age, 10 distributions with slightly different means and standard deviations were generated for each age category (example is see Fig. 3). From each distribution, 1,000 cells were sampled to create statistical samples.

Figure 2 Three patterns of change in the standard deviation of the normal distribution with aging are set up in the simulation analysis.

(A) Monotonic increase. (B) Non-monotonic. (C) Stable.

Figure 3 Example of the mean and standard deviation of the simulated sample distributions for each aging pattern of the gene.

Ten samples are simulated for each age category. For each gene, an association analysis between age and distribution was performed on a total of 40 samples.

In genes with monotonic parameter changes, consistently the highest correlation coefficients were observed (Fig. 4A). In the genes with non-monotonic changes, differences between groups were detected as moderate correlation coefficients (Fig. 4A). In the stable situation, the correlation coefficients were distributed around zero (Fig. 4A). While small Mantel test P-values are observed in the genes with monotonic increasing or non-monotonic parameter change, it is not in the gene with stable pattern (Fig. 4B). This result is consistent with the interpretation of the correlation coefficient results. Pseudo-bulk data analysis using the same dataset has shown that correlation coefficients near zero are observed in all cases (File S1). Pseudo-bulk expression values were defined as the average of the single-cell expression values in each sample. Conventional bulk gene expression analysis cannot directly capture such changes in variance because it is based on obtaining the average of the expression values of the cells in the sample. These results suggest that this method can detect the association between changes in the distribution of single-cell expression profiles and aging in a data-driven manner.

Figure 4 The result of simulation data analysis.

(A) The boxplot of Spearman’s correlation coefficient values for the simulated genes with each type of age-related pattern. (B) The boxplot of Mantel test P values for the simulated genes with each type of age-related pattern.

We performed the following computer experiments to verify the validity of this workflow. First, we performed an analysis using the Jensen–Shannon (JS) distance and the Kolmogorov–Smirnov (KS) distance instead of the Hellinger distance as a measure of the distance between probability distributions. JS distance is a symmetric distance measure based on KL divergence (Lin, 1991). In the calculation of KS distance, we applied the KS.diss functions in an R package “provenance” (Vermeesch, Resentini & Garzanti, 2016) to the two statistical samples. For both distance indices, we observed the same results as when using the Hellinger distance (File S2). The results have shown that the distance between probability distributions is interchangeable with other indices proposed in the field of information theory.

Real scRNA-seq data analysis

We applied the proposed workflow to the large-scale single-cell RNA-seq dataset for the aging study and presented the histograms of the correlation coefficients obtained for the four organs in Fig. 5. The shapes of the histograms were similar among the four organs, and genes with a correlation coefficient >0.85 cutoff value were considered as those showing changes in single-cell expression during aging (Table 1). In all tissues, the presence of genes with large positive correlation coefficients is observed. In limb muscle, 13 genes passed this threshold while no genes passed this criterion in other tissues.

Figure 5 Histograms of Spearman correlation coefficient and P value of Mantel test of association analysis between single cell expression profile and aging in kidney, limb muscle, lung, and marrow.

P values were under multiple adjustment by the BH method.

Table 1 Genes whose single cell expression profile changes with age in each tissue (Spearman’s correlation coefficien t > 0.85).

Tissue	Gene	Coefficient	
limb muscle	Jhdm1d	0.937	
limb muscle	Grb10	0.905	
limb muscle	Jmjd6	0.903	
limb muscle	Acot9	0.895	
limb muscle	Aldh2	0.889	
limb muscle	AI607873	0.886	
limb muscle	Plac9	0.886	
limb muscle	Gt(ROSA)26Sor	0.884	
limb muscle	Six1	0.864	
limb muscle	Wisp1	0.862	
limb muscle	Nr1d2	0.862	
limb muscle	Tnfsf9	0.858	
limb muscle	plscr1	0.855	

For genes detected by correlation coefficients, visualization of the shape of the distribution in a histogram or QQ plot would be useful for the biological interpretation of the results. Histograms and QQ plot visualization for all the genes listed in Table 1 are shown in Files S3 and S4, respectively. Even among those genes that are strongly associated with age and single-cell expression profile, we observe that the specific patterns of change are diverse.

The gene with the largest correlation coefficient was Jhdm1d. Jhdm1d, also known as KDM7A gene, is primarily known for its involvement in epigenetic regulation through histone demethylation, impacting gene expression and cellular processes which indirectly related to the aging process (Yang et al., 2019). Figures 6 and 7 are histograms and QQ plots of the single-cell expression profile of the Jhdm1d gene as an example of other distributional aging changes. The distributions are also observed as a mixed distribution of cells negative and positive for this gene. At 30 months of age, the peak of the distribution of Jhdm1d-positive cells are observed to shift to the right.

Figure 6 Histogram visualization of age-related changes in the single-cell expression profile of the Jhdm1d gene.

The x-axis represents RNA expression, whereas the y-axis represents the frequency. Cells from all samples in each age category are pooled.

Figure 7 QQ plot visualization of age-related changes in the single-cell expression profile of the Jhdm1d gene.

The x-axis represents the distribution at the youngest age and the y-axis represents the distribution at other ages. Cells from all samples in each age category are pooled.

As another example, the histograms and QQ plot of the single cell expression distribution of Aldh2 gene were present in Figs. 8 and 9, respectively. It is known that Aldh2 deficiency promotes age-related muscle loss, especially in oxidative fibers, which may be associated with an increased accumulation of oxidative stress via mitochondrial dysfunction (Kasai et al., 2022). The distributions are also observed as a mixed distribution of cells negative and positive for this gene. The number of cells in the bin with range between 0.5 to 1 decreases with age, suggesting that there may be a decrease in the number of intermediate cells between positive and negative cells. Screening using correlation coefficients, combined with visualization using histograms and QQ plots, allows us to examine changes in diverse single-cell expression profiles with aging.

Figure 8 Histogram visualization of age-related changes in the single-cell expression profile of the Aldh2 gene.

The x-axis represents RNA expression, whereas the y-axis represents the frequency. Cells from all samples in each age category are pooled.

Figure 9 Histogram visualization of age-related changes in the single-cell expression profile of the Aldh2 gene.

The x-axis represents the distribution at the youngest age and the y-axis represents the distribution at other ages. Cells from all samples in each age category are pooled.

Normally, changes in single-cell expression profiles are affected by both shifts in cell-level expression and changes in fractions of cellular subsets, but this method can detect them without distinguishing between them. If you want to focus only on shifts in cell-level expression, you can apply this method to a single cell type after cell type annotation with existing methods. We applied our workflow to each tissue and the major cell type. In this analysis, we used the following combinations of tissue and cell types as examples (Kidney: “B cell”, limb muscle: “mesenchymal stem cell”, Lung: “classical monocyte”, Marrow: “granulocyte”) as an example. Histograms of the correlation coefficients for each cell subset analysis are shown in File S5 . This analysis also suggested the presence of many genes associated with age. As with the whole tissue analysis, the age-related genes passing the threshold (0.85) have been observed only in limb muscle (21 genes, File S6). These genes were not consistent with the age-related genes in the analysis of whole tissue analysis. There are only two genes (Grb10, plac9) overlapped between two results.These results suggest that the detected genes will be different depending on the focus and range of cell populations.

Discussion

In this study, we proposed a novel workflow to detect changes in the distribution of any single-cell expression profile with aging and identified several genes whose single-cell expression profiles changed during aging. Although various patterns of differences in cell population profiles may occur with aging, this workflow can be used to screen for genes that are strongly associated with age, irrespective of the specific pattern. Indeed, visualizing a histogram of genes with high scores could offer what specific distributional changes are occurring to provide biological insights. This method serves as a natural extension of traditional bulk data analysis, which assesses the association between gene expression levels and age. In recent years, as more single-cell expression data are obtained from multiple samples, we anticipate that this method will be useful increasingly in future studies.

This study has the following limitations. First, this workflow can only be used to analyze genes that are expressed at levels that are sufficient for distance quantification of distribution. Consequently, the analysis of the real scRNA-seq dataset in this study was limited to the top 3,000 genes with the highest average expression levels. Second, this workflow yields only correlation coefficients to age; additional experiments, analysis, and literature reviews are required to gain further biological insights. Despite these limitations, the developed workflow is still expected to be useful as a straightforward method for screening genes in aging scRNA-seq datasets.

As a future perspective, this analytical framework is not limited to single-cell expression data and can be extended to the distribution of various cellular phenotypes as they evolve with age. For example, single-cell epigenomics analyses have shown significant age-related changes in aspects such as open chromatin profiles, histone modifications, and DNA methylation profiles (He et al., 2020; Zhang et al., 2022; Hernando-Herraez et al., 2019). It has also been suggested that cellular phenotypes such as cell morphology can be used as aging biomarkers (Phillip et al., 2017). Further, the framework used in this method can detect monotonic changes in distribution without any special assumptions. As such, it may serve as an initial screening approach for identifying novel factors related to aging.

Conclusion

In this study, we proposed a novel workflow to detect changes in the distribution of arbitrary monotonic age-related changes of single-cell expression profiles. It is suggested that this technique is potentially useful as a straightforward screening method to identify cellular features related to aging.

Supplemental Information

Supplemental Information 1 Box plots of Spearman correlation coefficients between pseudo-bulk expression values and age

Click here for additional data file.

Supplemental Information 2 Results of simulation analysis when using different probability distribution distance indices (Hellinger distance: HL, Jensen Shannon distance: JS, Kolmogorov–Smirnov: KS)

A box-and-whisker plot of the Spearman correlation coefficient and P-value for each aging pattern is described.

Click here for additional data file.

Supplemental Information 3 Histogram visualization of age-related changes in the single-cell expression profile of all detected genes

The x-axis represents the distribution at the youngest age and the y-axis represents the distribution at other ages. Cells from all samples in each age category are pooled.

Click here for additional data file.

Supplemental Information 4 QQ plot visualization of age-related changes in the single-cell expression profile of all detected genes

The x-axis represents the distribution at the youngest age and the y-axis represents the distribution at other ages. Cells from all samples in each age category are pooled.

Click here for additional data file.

Supplemental Information 5 Histograms of Spearman correlation coefficient and P value of Mantel test of association analysis between single cell expression profile and aging in B cell (kidney), mesenchymal stem cell (limb muscle), classical monocyte (Lung), granulocyte (marrow), res

Click here for additional data file.

Supplemental Information 6 Genes whose single cell expression profile changes with age in each tissue and cell type combinations (Spearman’s correlation coefficien t > 0.85)

Click here for additional data file.

We would like to thank FORTE Inc. for proofreading this manuscript.

Additional Information and Declarations

Competing Interests

Author Contributions

Data Availability

The authors declare there are no competing interests.

Jian Hao Cheng conceived and designed the experiments, performed the experiments, analyzed the data, prepared figures and/or tables, authored or reviewed drafts of the article, and approved the final draft.

Daigo Okada conceived and designed the experiments, authored or reviewed drafts of the article, and approved the final draft.

The following information was supplied regarding data availability:

The code used in this study are available at GitHub and Zenodo:

- https://github.com/chengjianhao123/scRNA/

- JIAN HAO, C., & OKADA, D. (2023). chengjianhao123/scRNA: Data-driven detection of age-related arbitrary monotonic changes in single-cell gene expression distributions. (single_cell_RNA). Zenodo. https://doi.org/10.5281/zenodo.10396488.

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
