# Peer review of "Data-driven detection of age-related arbitrary monotonic changes in single-cell gene expression distributions"

_PeerJ, doi:10.7717/peerj.16851_

## Round 0.1 · original submission · Major Revisions

Please address the issues pointed out by all reviewers and amend the manuscript accordingly.

·

Basic reporting

Authors introduce a worlflow to detect distribution changes via association analysis in single-cell expression profiles among people with montonic age-related changes. In general, the story is simple and clear, references are sufficient, and figures are informative.

Also there are some format errors but easy to fix e.g. wrong space position in line 83,84, words error in line 108, and invaild github link in line 159.

Experimental design

Questions & comments

1. Methology
a. Why KS statistics? KS statisctis is calculated as the maximum difference between 2 distributions which could be sensitive to outliers. Mean or median difference could be more robust.

b. Why Spearman's correlation? Spearman's correlation is for oridinal (ranked variarles). Difference in KS statistics could be considerable and the gap between ages can be heterogeneous.

c. Most important one, KS statistics are commutable while subtraction is not commutable, which means there could be a sign flip if you change the order of two samples. Consequently, the final coefficient could be filped.

2. Simulation part
As mentioned above, Spearman's correlation is for oridinal (ranked variarles). Why do you consider normal distribution in the simulation?

3. Real data analysics
a. Sample size (around 10) is too small regarding different ages. In average, just 2 mice in one age bin.

b. A significant advantage of singe-cell RNA sequence compared to bulk-leve is the ample infomation about the cell type, which is not considered in your report. Just 4 organs are selected. Or the bulk level data seems to be more appropriate.

c. Why do you set the coefficient threshold to be 0.4? More interpretation needed.

Validity of the findings

Overall, the framework looks too simple to be novel, I don't think it goona work in the real application. Authors need show either more advanced statistical methodology or evidence within more datasets.

Reviewer 2 ·

Basic reporting

Cheng et al propose a new approach based on Kolmogorov-Smirnov distance and spearman’s correlation to detect monotonic changes in single cell expression profile. The authors demonstrate their workflow via a simulation setting and a public aging mouse scRNA-seq dataset.

1. Please provide a comprehensive overview of different distance metrics (or divergence measures), including both parametric and non-parametric distance metrics. It’d be great to list the advantages and disadvantages of each method. Please point out the rationale of using KS distance in this setting.

2. The authors fail to account for different cell types in the single-cell settings, in both simulation and real data application. For each sample, different cells may come from different cell states, different cell states may have different association with age. Would suggest the authors to follow the single-cell analysis pipeline (e.g., data integration, cell type annotation, etc) rather than treat all cells the same.

Experimental design

1. Please work on the Methods section to provide a clear and concise description of the simulation data generation (e.g., use bullet points and formulas). It’s hard to capture the key steps of how the data are generated. Also, please provide a setting that can mimic the real-word single-cell data (e.g. at least 10k cells, increase number of permutations in the null construction).

2. Please add more simulation settings to demonstrate the robustness and validity of the method. For example, here are some factors that could be considered but not limited to:
a. Non-normal distribution (e.g. Poisson or negative binomial)
b. Split the features into three groups: monotonic increasing, decreasing and others (e.g. constant, non-monotonic change (first increase then decrease), etc) . This will serve as gold standard in the simulation settings. Please provide the expression pattern of the truth in main figure (e.g. using heatmap).
c. Include different cell types with different expression changing pattern.
d. Compare each simulation result with the ground truth in main or supplemental figures.

3. Please use a rigorous statistical test (e.g., permutation-based) to detect the statistically significant age-related genes and visualize them in boxplot or heatmap in the supplemental figure. Please consider using multiple testing adjustment.

4. In real data application, using a threshold of 0.4 is quite subjective and arbitrary. It is not clear whether a null distribution is used in the Methods and data application. For example, in Table 1 and Fig3, a statistical test should be conducted for each gene and provide a p-value and FDR for multiple testing adjustment. If a null will be computed, please include this step in Fig1 workflow.

5. For Fig4, it’s hard to observe the expression pattern changes with age. Please provide an alternative plot (e.g., boxplot) to demonstrate the aging-related gene expression changes. The x axis could be the age group, and y axis could be normalized gene expression. Also, please include all monotonic changing genes detected by the proposed method.

Validity of the findings

1. In real data application, the authors only focused on a subset of 100 genes with highest average expression. Please provide a comprehensive analysis using all genes and follow the single-cell analysis pipeline.

2. Please provide the runtime and memory usage in practice. The number of cells could range from thousands to millions of cells, depending on the study design.

3. A more direct approach is to compute pseudobulk gene expression profile for each sample in each cell type, then a linear regression model can be applied to detect the association between sample’s phenotype (e.g. age) and pseudobulk expression profile. P-value and FDR are computed, and genes can be ordered and detected by coefficients and FDR. Please provide a direct comparison between the proposed method and this regression approach.

4. Please conduct a more thorough literature search to support of the findings. Please include both tissue and cell type information.

5. Please include a second single-cell dataset (preferably a human dataset) to demonstrate the proposed methods.

·

Basic reporting

Cheng and Okada pose an interesting question (how can we identify genes that change with age?) and present a simple method to answer it in a manner that differs from commonly used approaches. The manuscript is well written and easy to read, provides all necessary code to recreate the analysis, and could be applicable to other time-course scRNA-seq studies beyond those focused solely on aging. However, there are some a few concerns that I have with the manuscript that I feel need to be addressed before publication in order to make the method sufficiently accessible and reusable to readers.

Experimental design

Major Comments
- The authors used real time course aging scRNA-seq data from The Tabula Muris Consortium, but do not comment on how their new new method compares to the original author’s analysis. In their 2020 Nature paper, the Tabula Muris Consortium modeled time as a continuous variable to identify the most significantly changing genes. None of the genes that are listed in table 1 were identified in the top 40 differentially expressed genes for the three tissues (kidney, lung, limb muscle) in the original Tabula Muris Consortium paper. Why do the authors think that is? The authors have an opportunity to provide an example of how their method offers a different insight that differs from the benchmark. A detailed comparison of the top genes across the common tissue types identified by the original paper and the new method presented by these authors would be very helpful – answering questions such as whether there any overlapping genes in the ranked lists or whether the genes differ but share pathway functionality or ontology would provide the reader with ample context.
- The authors selected the top 100 genes by expression, but this seems to be an arbitrary number. I suggest that they perform a sensitivity analysis to determine a minimum count threshold that their method is suitable for you. The top 100 genes by expression will vary from scRNA-seq study to study, so a more robust threshold is needed for readers of the manuscript can properly use the method in their own analysis.
- Table 1 shows the genes with a correlation coefficient > 0.4, why was this number selected? I would recommend showing substantially more genes for each tissue type and show the results for all tissue types. It leaves the reader wondering whether there are many genes just below the cutoff threshold. A table could still be used for top genes, and allow the reader to determine whether any top genes are common amongst different tissues, or some sort of plot including correlation and expression through the aging timepoints could also be useful.

Validity of the findings

Major Comments
- scRNA-seq data preprocessing: the authors do not describe how the scRNA-seq data was processed in their workflow. This is essential to describe for reuse of their method. Do they assume that the scRNA-seq counts are normally distributed, or do they apply a variance stabilization method or log transformation of counts prior to the Kolmogorov-Smirnov dissimilarity test? I appreciate that the authors have included all code on GitHub, however the methods section needs a more detailed description of the specific steps in the pipeline. There is only a proposed workflow and methods of the simulated data, but not the specific steps for the methods of the calculations themselves.

Additional comments

Minor Comments
- Mouse gene naming convention should be used (italicized lowercase, not uppercase like human genes or non-italics implying protein)
- Table 1, it should be Myl6 not My16
- How do these highly correlated genes change in expression with time? See last major comment for possible areas of expansion.
- Introduction is easy to read and gives the reader context, however a paragraph about alternative ways to identify genes that significantly change through a scRNA-seq time course (aging or otherwise) would be very useful.

---

## Round 0.2 · accepted · Accept

All concerns of the reviewers were adequately addressed and revised manuscript is acceptable now.

·

Basic reporting

The authors updated the manuscript regarding my concerns. No further change is needed.

Experimental design

The authors updated the manuscript regarding my concerns. No further change is needed.

Validity of the findings

The authors updated the manuscript regarding my concerns. No further change is needed.

Reviewer 2 ·

Basic reporting

no comment

Experimental design

no comment

Validity of the findings

no comment